# ThanoRA: Task Heterogeneity-Aware Multi-Task Low-Rank Adaptation

## Abstract

Low-Rank Adaptation (LoRA) is widely adopted for downstream fine-tuning of foundation models due to its efficiency and zero additional inference cost. Many real-world applications require foundation models to specialize in several specific tasks simultaneously, motivating the need for efficient multi-task downstream adaptation. To address this need, existing studies have primarily explored two directions: *Model Merging with LoRA*, which shows advantages in training-free scenarios but still lags behind multi-task training in overall performance; and *MoE-based LoRA* approaches, which improve multi-task learning performance but introduce routers that hinder the mergeability of LoRA parameters and incur considerable inference overhead, thereby limiting real-world deployment practicality. To this end, we propose **ThanoRA**, a Task Heterogeneity-Aware Multi-Task Low-Rank Adaptation framework that enables effective, efficient and unified multi-task downstream adaptation without introducing additional structure. ThanoRA performs multi-task learning by tailoring subspace allocation at initialization and enforcing diversity preservation throughout training: it allocates varying dimensions to construct task-specific low-rank subspaces driven by inter-task heterogeneity, enabling fine-grained knowledge injection, while diversity-preserving regularization mitigates task interference and subspace collapse, thereby fully exploiting the low-rank capacity. Extensive experiments across multimodal and text-only benchmarks under varying multi-task mixtures demonstrate that ThanoRA consistently outperforms strong baselines, surpassing even separate task-specific fine-tuning, while introducing no additional structures or inference overhead.

## 1 Introduction

With the rapid development of foundation models Bai et al. (2025); Wang et al. (2025); Grattafiori et al. (2024); Li et al. (2024a); Liu et al. (2023), their general capabilities across a wide range of tasks have been significantly enhanced. Nevertheless, they often require downstream adaptation to perform well on specific target tasks Hu et al. (2022); Huang et al. (2025). To address this, a variety of parameter-efficient fine-tuning (PEFT) methods have been proposed to adapt foundation models with minimal computational and storage overhead Houlsby et al. (2019); Lester et al. (2021); Liu et al. (2022); Li & Liang (2021); Hu et al. (2022). Among them, LoRA Hu et al. (2022) and its variants Meng et al. (2024); Liu et al. (2024b); Yang et al. (2024); Hayou et al. (2024) are particularly appealing due to their low training cost and the ability to merge adapted parameters into the base model without incurring inference overhead. Although PEFT methods such as LoRA have shown competitive performance, recent studies suggest that they may encounter challenges on heterogeneous corpora involving multiple tasks or domains, where task conflicts and interference can hinder adaptation Tian et al. (2024). Meanwhile, real-world applications often require a single model to acquire several targeted capabilities simultaneously Li et al. (2024b); Liu et al. (2024a), motivating the need for efficient multi-task downstream adaptation.

Existing efforts toward multi-task efficient adaptation have primarily explored two directions. The first is *Model Merging with LoRA* (see fig. 1.a), where individually fine-tuned LoRA modules from different tasks are directly merged to form a multi-task model Stoica et al. (2025); Zeng et al. (2025); Zheng et al. (2025). This approach is attractive due to its training-free nature in combining arbitrary sets of tasks, but the lack of a shared optimization process limits its ability to resolve inter-task conflicts, resulting in **inferior performance to multi-task training**. The second line of research

*combines LoRA with Mixture-of-Experts (MoE)* (see fig. 1.b) architectures Li et al. (2024b); Shen et al. (2024); Zhao et al. (2025); Tian et al. (2024); Chen et al. (2024); Wu et al. (2024b). These methods employ routers to dynamically dispatch inputs to different LoRA experts, thereby alleviating conflicts and improving multi-task performance. Nevertheless, the reliance on routers prevents the adapted parameters from being merged back into the base model, leading to **considerable inference overhead** and **extra storage requirements**, thereby hindering real-world deployment.

To this end, we aim to develop an efficient multi-task low-rank adaptation framework better aligned with real-world deployment, enabling large foundation models to master the combination of targeted tasks without incurring additional inference or storage overhead. This raises the following challenge:

**Challenge I)** *How to effectively inject multi-task knowledge into the low-rank subspace?*

Conducting multi-task training in a unified LoRA framework without additional non-mergeable structures requires tasks to share the same trainable parameter subspace, often leading to conflicts and performance degradation. Thus, the key challenge is to compress multi-task knowledge into a shared low-rank space. To address this, we first note the following fact:

*Tasks exhibit heterogeneous representational demands across layers of large foundation models.*

Prior studies have shown that LoRA modules exhibit varying representational requirements across layers and modules, necessitating different rank configurations Zhang et al. (2023). In the context of multi-task learning, we further extend this conclusion by observing that different tasks present heterogeneous representational demands across different layers. As illustrated in fig. 1.d, consider a concrete example: some tasks involve relatively simple visual content but require complex textual reasoning, whereas others display the opposite pattern. This heterogeneity results in divergent representational capacities required by the foundation model at different layers, as reflected in the accompanying bar chart. While this example illustrates visual–textual heterogeneity, real tasks exhibit multidimensional variations that further accentuate representational differences. Consequently, applying a uniform adaptation strategy in multi-task training may fail to capture these complexity differences, leading to suboptimal resource allocation. This motivates us to **allocate low-rank resources in a task-aware manner, aligning the representational capacity at each layer with the heterogeneous demands of different tasks.** To this end, we propose **Heterogeneity-Aware Subspace Initialization (HASI)**. Specifically, task-specific priors are extracted based on instruction-previewed SVD, as similarly employed in Yang et al. (2024). We then estimate task heterogeneity via the spectral entropy of singular values, which serves as a proxy for the subspace complexity required by each task. Based on the relative heterogeneity of each task, we partition the $R$-dimensional subspace spanned by the LoRA modules at each layer. Each task-specific subspace is then initialized with the truncated singular vectors corresponding to its allocated dimension, enabling fine-grained, layer-wise knowledge injection tailored to task complexity.

While HASI enables a proper allocation of low-rank subspaces and task-aware knowledge injection at initialization, during multi-task training the parameter subspaces of different tasks still share gradient update signals. As a result, the initially disentangled task-specific subspaces may gradually lose their diversity, leading to *subspace collapse* (see Sec. 3.2.2 for definition). Thus, this raises the second key challenge:

**Challenge II)** *How to mitigate low-rank subspace collapse during multi-task training?*

To mitigate this, we further propose **Subspace-Preserving Regularization (SPR)**, which enforces orthogonality among different task-specific subspaces during training while retaining a partially shared collaborative subspace for knowledge sharing. This orthogonality constraint enhances the expressiveness of the low-rank subspaces initialized by HASI and effectively alleviates the problem of subspace collapse (refer to Sec. 4.3.3 for experimental evidence).

Notably, HASI and SPR work *synergistically*: HASI creates heterogeneous subspaces for task-aware knowledge injection *at initialization*, while SPR preserves their diversity *throughout training*, jointly enabling effective multi-task adaptation within a unified, mergeable LoRA framework. Our contributions are summarized as follows:

❶ *Revealing and Modeling Heterogeneous Representational Demands Across Layers.* We reveal that downstream tasks vary in complexity and characteristics across layers, motivating layer-wise heterogeneity modeling within a unified framework.

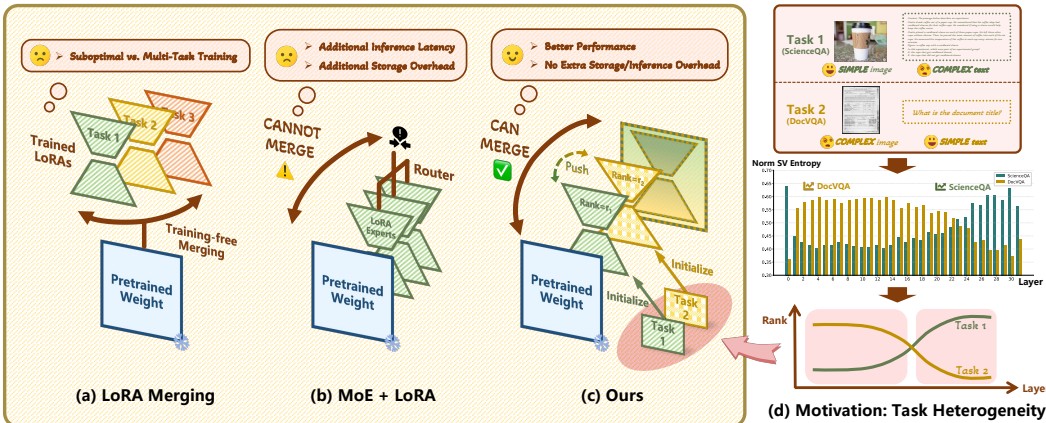

Figure 1: **Comparison and Motivation.** (a) LoRA Merging acquires multi-task knowledge without training but remains suboptimal compared to multi-task training. (b) MoE-based LoRA mitigates task conflicts via routers but compromises parameter mergeability. (c) Our method enables efficient multi-task adaptation through mergeable LoRA with task-specific subspace modeling. (d) Tasks show heterogeneous complexity across layers, motivating the allocation of task-specific subspaces to better match their representational demands.

❷ *Effective and Efficient Multi-Task Low-Rank Adaptation Framework.* We propose ThanoRA to address the challenges of task heterogeneity modeling and subspace collapse during multi-task training, enabling effective multi-task adaptation while preserving parameter mergeability and inference efficiency, supplementing the shortcomings of merging- and MoE-based methods.

❸ *Comprehensive Empirical Validation.* We conduct extensive experiments on multimodal and text-only benchmarks, consistently demonstrating robust performance improvements and efficiency advantages over strong baselines, surpassing even separate task-specific fine-tuning.

## 2 RELATED WORK

### 2.1 PARAMETER-EFFICIENT FINE-TUNING.

Parameter-Efficient Fine-Tuning (PEFT) methods aim to adapt large foundation models with minimal trainable parameters, avoiding the prohibitive cost of full fine-tuning Houlsby et al. (2019); Lester et al. (2021); Li & Liang (2021); Liu et al. (2022); Pfeiffer et al. (2020); Karimi Mahabadi et al. (2021); Wu et al. (2024a); Hu et al. (2022); Liang et al. (2025). Among these, Low-Rank Adaptation (LoRA) Hu et al. (2022) is widely adopted due to its simplicity and zero additional inference cost, injecting trainable low-rank matrices into existing weight layers without modifying the model architecture. Numerous extensions have been proposed to improve its performance. DoRA Liu et al. (2024b) decomposes pretrained weights into magnitude and direction components to improve stability and adaptation capacity. PiSSA Meng et al. (2024) accelerates convergence by initializing low-rank matrices with the principal components of the pretrained weights instead of random noise. Other approaches include dynamic rank allocation Zhang et al. (2023), vector-based random matrix adaptation Kopiczko et al. (2023), instruction-previewed initialization Yang et al. (2024). These variants improve the capacity and convergence behavior of LoRA, without sacrificing its efficiency in both training and inference.

### 2.2 MULTI-TASK LEARNING.

Multi-task learning (MTL) aims to enhance generalization by jointly training on multiple tasks, enabling shared representations to improve knowledge integration Crawshaw (2020); Zhang & Yang (2021); Ban & Ji (2024); Agiza et al. (2024); Huang et al. (2023). Classical approaches include parameter sharing Wallingford et al. (2022); Ma et al. (2019), task-specific structure Guo et al. (2020); Sun et al. (2021), or regularization-based techniques Zhang & Yeung (2014); Yang & Hospedales (2016). With the rise of large foundation models, efficiently adapting them to diverse downstream tasks has become a central focus in recent MTL research. Recent studies on parameter-efficient

multi-task adaptation can be broadly divided into two categories: **(i) Model Merging with LoRA.** These methods directly merge LoRA modules trained on individual tasks Stoica et al. (2025); Zeng et al. (2025); Zheng et al. (2025). While attractive for training-free scenarios, they generally lag behind multi-task training due to the absence of explicit inter-task optimization. **(ii) MoE-based LoRA.** A growing number of works combine LoRA with Mixture-of-Experts (MoE) architectures Li et al. (2024b); Shen et al. (2024); Zhao et al. (2025); Tian et al. (2024); Chen et al. (2024); Wu et al. (2024b); Feng et al. (2024); Wang et al. (2024); Zhu et al. (2025). For example, MixLoRA Shen et al. (2024) mitigates task interference by conditionally constructing task-aware LoRA factors for task-specific routing, while HydraLoRA Tian et al. (2024) introduces asymmetric experts to improve multi-task adaptability. However, these approaches rely on non-mergeable routing structures, incurring considerable inference and additional storage overhead, thereby undermining the deployment efficiency of LoRA. While LoRA model merging sacrifices overall performance and MoE-based LoRA designs compromise mergeability and efficiency, neither fully addresses the need for effective and practical multi-task adaptation.

## 3 METHODOLOGY

### 3.1 PRELIMINARY

**Low-Rank Adaptation.** LoRA introduces trainable low-rank matrices into pretrained models to achieve efficient fine-tuning with minimal parameter overhead. Specifically, for a linear projection $W \in \mathbb{R}^{d_{\text{out}} \times d_{\text{in}}}$, LoRA reparameterizes it as:

$$W_{\text{LoRA}} = W + \Delta W = W + BA, \tag{1}$$

where $A \in \mathbb{R}^{r \times d_{\text{in}}}$, $B \in \mathbb{R}^{d_{\text{out}} \times r}$, and $r \ll \min(d_{\text{in}}, d_{\text{out}})$ is the rank of LoRA. This not only minimizes training overhead but also allows the updated weights to be seamlessly merged into the original model after training, which is a key advantage of LoRA for deployment efficiency.

**Instruction-Previewed Initialization.** Following previous work Yang et al. (2024), we extract task-specific priors by forwarding a small batch (e.g., 500) of task-specific samples through the pretrained foundation model. For each linear layer, we collect the input activation $X \in \mathbb{R}^{d_{\text{in}} \times BL}$, compute the covariance $C = XX^T$, and derive a task-aware Singular Value Decomposition (SVD) as follows:

$$\widehat{W} = \text{SVD}(WC)C^{-1} = \sum_{i=1}^{R} \sigma_i \mathbf{u}_i \hat{\mathbf{v}}_i^\top, \tag{2}$$

yielding singular values $\{\sigma_i\}$ and corresponding vectors $\mathbf{u}_i, \hat{\mathbf{v}}_i$ for knowledge extraction.

### 3.2 PROPOSED METHOD

To address the challenges of incorporating heterogeneous task knowledge into the low-rank subspace (Challenge **I**) and mitigating subspace collapse (Challenge **II**) in multi-task low-rank adaptation, we propose **ThanoRA**, a Task Heterogeneity-Aware framework that enables effective and efficient multi-task adaptation.

#### 3.2.1 HETEROGENEITY-AWARE SUBSPACE INITIALIZATION (HASI)

**Motivation.** *Real-world tasks often exhibit heterogeneity across various aspects, leading to divergent representational demands across layers of large foundation model.* For example, as shown in fig. 1(c), in multi-task fine-tuning of MLLM, DocVQA Mathew et al. (2021) task presents visually complex but textually simple inputs, whereas ScienceQA Lu et al. (2022) task features the opposite. This heterogeneity is further reflected in the normalized spectral entropy curves (computed via Eq. equation 3), where each task shows distinct complexity patterns across layers. These observations align with prior findings that early layers of MLLMs focus on cross-modal fusion, while later layers predominantly process the fused semantic representations Huang et al. (2025); Zhang et al. (2024). Notably, this example reflects only a single aspect of heterogeneity, while real-world tasks involve richer and multidimensional forms of variation. Such observations highlight the necessity of explicitly modeling heterogeneity in multi-task adaptation.

To capture such discrepancies, we construct task-specific low-rank subspaces aligned with layer-wise representational heterogeneity. We first obtain a *task-aware representation matrix* $\widehat{W} \in$

Figure 2: **Overview of ThanoRA.** *(a) Heterogeneity-Aware Subspace Initialization* via entropy-guided rank allocation; and *(b) Subspace-Preserving Regularization* with orthogonality constraints, which together enable multi-ability acquisition across downstream tasks. Please refer to Sec. 3.2.

$\mathbb{R}^{d_{\text{out}} \times d_{\text{in}}}$ using instruction-previewed SVD, as defined in eq. (2). This captures the dominant semantic directions associated with the task-specific response to the pretrained model. Then, to quantify the complexity of the representation required by each task at a given layer, we compute the *spectral entropy* of the singular values $\boldsymbol{\sigma} \in \mathbb{R}^{r_{\text{total}}}$ of $\widehat{W}_t$ for task $t$ as:

$$\mathcal{H}(\boldsymbol{\sigma}) = -\sum_{i=1}^{R} p_i \log(p_i), \quad p_i = \frac{\sigma_i}{\sum_j \sigma_j}, \tag{3}$$

where $r_{\text{total}}$ denotes the total dimension budget for truncating the SVD of $\widehat{W}_t$. This quantifies the representational complexity of the subspace at each layer for a given task.

Furthermore, we compress multi-task knowledge into a shared low-rank space, which in LoRA is naturally constrained by the upper bound $r_{\text{total}}$. Under this fixed total dimension budget $r_{\text{total}}$, we allocate a portion $r_t^\ell$ of the LoRA rank to each task $t$ at layer $\ell$ proportionally to the normalized entropy across tasks:

$$r_t^\ell \propto \text{Softmax}\left(\frac{\mathcal{H}_t^\ell}{\tau}\right), \tag{4}$$

where $\tau$ is a temperature hyperparameter controlling the sharpness of the allocation. To promote task cooperation, we reserve an additional cooperative subspace with rank $r_{\text{coop}} = \left\lfloor \frac{r_{\text{total}}}{T+1} \right\rfloor$, where $T$ denotes the number of tasks.

The remaining task-specific dimension budget $r_{\text{task}} = r_{\text{total}} - r_{\text{coop}}$ is then distributed by discretizing the soft allocations:

$$r_t^\ell = \max\left(r_{\min}, \left\lfloor r_{\text{task}} \cdot \frac{\exp(\mathcal{H}_t^\ell/\tau)}{\sum_{j=1}^{T} \exp(\mathcal{H}_j^\ell/\tau)} \right\rfloor\right), \tag{5}$$

where $r_{\min}$ ensures each task receives a minimum allocation to prevent vanishing ranks for certain tasks.

For each task $t$ and layer $\ell$, let $(\mathbf{U}_t^\ell, \mathbf{V}_t^\ell, \boldsymbol{\Sigma}_t^\ell)$ be the truncated SVD components of the instruction-previewed weight estimation. We construct the initial LoRA matrices of each task by scaling the singular vectors with the square roots of their corresponding singular values:

$$\mathbf{B}_t^\ell = \mathbf{U}_t^\ell \cdot \left(\boldsymbol{\Sigma}_t^\ell\right)^{\frac{1}{2}}, \quad \mathbf{A}_t^\ell = \left(\boldsymbol{\Sigma}_t^\ell\right)^{\frac{1}{2}} \cdot \mathbf{V}_t^\ell. \tag{6}$$

Having obtained the task-specific components $\mathbf{B}_t^\ell$ and $\mathbf{A}_t^\ell$ for each task $t$, we define the corresponding low-rank initialization subspace at layer $\ell$ as $\mathbf{W}_t^\ell \triangleq \mathbf{B}_t^\ell \mathbf{A}_t^\ell$. We aim to fuse the initialization subspaces of different tasks. To this end, we realize the fusion via the following blockwise concatenation:

$$\mathbf{B}^\ell = \left[\mathbf{B}_1^\ell, \cdots, \mathbf{B}_T^\ell\right], \mathbf{A}^\ell = \left[\mathbf{A}_1^\ell; \cdots; \mathbf{A}_T^\ell\right], \tag{7}$$

We demonstrate in the proposition below that this blockwise concatenation is effect-equivalent to the additive composition of the task-specific low-rank components.

**Proposition 3.1** (Blockwise composition). *Let $\mathbf{B}_t^\ell \in \mathbb{R}^{d_{\text{out}} \times r_t}$ and $\mathbf{A}_t^\ell \in \mathbb{R}^{r_t \times d_{\text{in}}}$ for tasks $t = 1, \ldots, T$, and define $\mathbf{B}^\ell = [\mathbf{B}_1^\ell, \ldots, \mathbf{B}_T^\ell] \in \mathbb{R}^{d_{\text{out}} \times r}$ and $\mathbf{A}^\ell = [\mathbf{A}_1^\ell; \ldots; \mathbf{A}_T^\ell] \in \mathbb{R}^{r \times d_{\text{in}}}$ with $r = \sum_t r_t$. Then, for any input $\mathbf{X} \in \mathbb{R}^{d_{\text{in}} \times n}$,*

$$\mathbf{B}^\ell \mathbf{A}^\ell \mathbf{X} = \sum_{t=1}^{T} \mathbf{B}_t^\ell \mathbf{A}_t^\ell \mathbf{X}. \tag{8}$$

*Proof.* See Appendix C.1. □

Thus, we fuse task-specific LoRA initialization subspaces from different tasks with heterogeneous ranks via blockwise concatenation, enabling knowledge injection into low-rank subspaces. Meanwhile, this property ***preserves the initialization parameter subspace of each task explicitly***, facilitating the regularization of subsequent multi-task training. We then concatenate all task-specific LoRA components along with the cooperative block:

$$\mathbf{B}^\ell = \left[\mathbf{B}_1^\ell, \cdots, \mathbf{B}_T^\ell, \mathbf{B}_{\text{coop}}^\ell\right], \mathbf{A}^\ell = \left[\mathbf{A}_1^\ell; \cdots; \mathbf{A}_T^\ell; \mathbf{A}_{\text{coop}}^\ell\right], \tag{9}$$

where $r_{\text{coop}} = \left\lfloor \frac{r_{\text{total}}}{T+1} \right\rfloor$, $\mathbf{A}_{\text{coop}}^\ell \in \mathbb{R}^{r_{\text{coop}} \times d_{\text{in}}}$ is randomly initialized using Kaiming uniform distribution and $\mathbf{B}_{\text{coop}}^\ell \in \mathbb{R}^{d_{\text{out}} \times r_{\text{coop}}}$ is initialized as zero. All components are further scaled by a global factor $\gamma$ to modulate the strength of knowledge injection. Thus, we establish a task-aware, layer-specific low-rank parameterization that encodes heterogeneous priors into structurally decoupled subspaces, providing a strong starting point for collaborative multi-task adaptation.

### 3.2.2 SUBSPACE-PRESERVING REGULARIZATION (SPR)

**Motivation.** *While HASI in Sec. 3.2.1 ensures task-specific subspace initialization, receiving identical sample signals within the same low-rank subspace during multi-task training can cause overlaps in their representation directions, thereby eroding the diversity of the initialized subspaces.* This results in an under-utilization of the low-rank parameter space, where performance fail to improve with larger ranks (see fig. 5). We define this phenomenon as ***subspace collapse***. To address this, we aim to promote the diversity of the subspaces initialized by HASI during multi-task training.

**Decomposed Orthogonality.** However, directly imposing orthogonality constraints on the task-specific parameter subspaces $\mathbf{B}_t^\ell \mathbf{A}_t^\ell$ incurs substantial computational and memory overhead, as it requires computing pairwise Frobenius inner products between the full update matrices at every layer, and the cost grows rapidly with the number of tasks. Please see Appendix B for additional discussion. To mitigate this, we adopt a decomposed formulation by separately enforcing orthogonality on the LoRA factors $\mathbf{A}_t^\ell$ and $\mathbf{B}_t^\ell$, which serves as a sufficient condition for orthogonality of the full updates. This design significantly reduces the regularization overhead while maintaining effective subspace separation. We formalize this condition with the following proposition.

**Proposition 3.2** (Sufficient Condition for Orthogonality of LoRA Updates). *Let $\mathbf{W}_1 = \mathbf{B}_1\mathbf{A}_1 \in \mathbb{R}^{d_{\text{out}} \times d_{\text{in}}}$ and $\mathbf{W}_2 = \mathbf{B}_2\mathbf{A}_2 \in \mathbb{R}^{d_{\text{out}} \times d_{\text{in}}}$. If either*

$$\mathbf{B}_1^\top \mathbf{B}_2 = \mathbf{0} \quad \textit{or} \quad \mathbf{A}_1 \mathbf{A}_2^\top = \mathbf{0}, \tag{10}$$

*then $\langle \mathbf{W}_1, \mathbf{W}_2 \rangle_F = 0$.*

**Proof.** *See Appendix C.2* □

To enforce this principle during training, we introduce a subspace-diversifying regularization term that explicitly encourages diversity and prevents representation overlap between subspaces, thereby improving the utilization of the low-rank parameter space and mitigating subspace collapse. For each layer $\ell$, the regularization loss is defined as:

$$\mathcal{L}_{\text{SPR}}^\ell = \sum_{t_1 < t_2} \left\| \mathbf{B}_{t_1}^{\ell\top} \mathbf{B}_{t_2}^\ell \right\|_F^2 + \left\| \mathbf{A}_{t_1}^\ell \mathbf{A}_{t_2}^{\ell\top} \right\|_F^2, \tag{11}$$

where the pairwise Frobenius norms measure the degree of inter-task subspace entanglement. Minimizing this term encourages representational diversity across tasks, thereby preserving the decoupled subspace structure established during initialization.

During training, this regularization is applied jointly with the supervised objective. The total training objective becomes:

Table 1: **Comparison on two-task mixtures** across various multimodal and text-only datasets. *Merge* indicates whether the method can be merged back into original model for inference.

| Methods | IconQA - ScienceQA | | | | ChartQA - DocVQA | | | | CSQA - OBQA | | | | Merge |
|---|---|---|---|---|---|---|---|---|---|---|---|---|---|
| | IconQA | ScienceQA | Avg | Norm | ChartQA | DocVQA | Avg | Norm | CSQA | OBQA | Avg | Norm | |
| *Mixture of 2 Tasks* | | | | | | | | | | | | | |
| Individual FT | 77.53 | 87.51 | 82.52 | 100 | 33.00 | 30.64 | 31.82 | 100 | 74.53 | 76.20 | 75.37 | 100 | - |
| LoRA | 77.47 | 87.61 | 82.54 | 100.02 | 32.52 | 30.38 | 31.45 | 98.84 | 74.86 | 78.80 | 76.83 | 101.94 | ✓ |
| DoRA | 77.53 | 88.50 | 83.02 | 100.61 | 32.24 | 30.25 | 31.25 | 98.21 | 75.76 | 79.60 | 77.68 | 103.06 | ✓ |
| CorDA | 73.84 | 86.66 | 80.25 | 97.25 | 32.32 | 31.11 | 31.72 | 99.69 | 76.25 | 78.20 | 77.23 | 102.47 | ✓ |
| DARE | 70.33 | 84.10 | 77.22 | 93.58 | 27.36 | 27.91 | 27.64 | 86.86 | 72.56 | 70.00 | 71.28 | 94.57 | ✓ |
| KnOTS | 70.52 | 84.18 | 77.35 | 93.73 | 27.42 | 28.32 | 27.87 | 87.59 | 72.65 | 70.20 | 71.43 | 94.77 | ✓ |
| HydraLoRA$_{(3B1A)}$ | 76.88 | 87.80 | 82.34 | 99.78 | 32.44 | 31.09 | 31.77 | 99.84 | 74.61 | 77.60 | 76.11 | 100.98 | ✗ |
| HydraLoRA$_{(4B1A)}$ | 77.07 | 88.00 | 82.54 | 100.02 | 32.36 | 31.15 | 31.76 | 99.81 | 73.71 | 75.80 | 74.76 | 99.19 | ✗ |
| ThanoRA | 79.08 | 89.74 | **84.41** | **102.29** | 33.20 | 31.23 | **32.22** | **101.26** | 76.99 | 80.20 | **78.60** | **104.29** | ✓ |

$$\mathcal{L}_{\text{total}} = \mathcal{L}_{\text{task}} + \lambda \sum_{\ell} \mathcal{L}_{\text{SPR}}^{\ell}, \tag{12}$$

where $\lambda$ is a hyperparameter controlling the strength of subspace regularization. This formulation allows the model to jointly optimize task performance and representational diversity, ensuring that the subspace knowledge injection introduced by HASI remains effective throughout training.

# 4 EXPERIMENTS

## 4.1 EXPERIMENTAL SETUP

**Datasets and Architecture.** We evaluate our framework on multi-task settings covering both *multimodal* and *text-only* datasets, including three two-task mixtures (*IconQA + ScienceQA*, *DocVQA + ChartQA*, *CSQA + OBQA*) and a four-task mixture (*IconQA*, *ScienceQA*, *DocVQA*, *ChartQA*). Each dataset contributes 5,000 training samples. Detailed descriptions are provided in Appendix A.

**Implementation Details.** All baselines are implemented within the LLaVA-1.5[1] framework. All PEFT methods are applied to every layer of the LLM component in LLaVA. The learning rate is set to $2e-5$ and the global batch size to 32. We train for 3 epochs on multimodal datasets and 1 epoch on the text-only mixture. Following previous work Yang et al. (2024), we sample a small part of samples (e.g. 500 per task) to initialize the task-specific subspaces in HASI.

**Counterparts.** We compare ThanoRA against three representative categories of methods: ❶ *Mergeable LoRA Variants* , ❷ *Model Merging*, and ❸ *MoE-based LoRA*, including: (a) **Individual Fine-tune**, where a separate LoRA module is trained for each task individually, serving as a strong baseline for multi-task adaptation; (b) **LoRA** Hu et al. (2022), which updates parameters in a low-rank subspace for efficient adaptation; (c) **DoRA** Liu et al. (2024b), which decou-

Table 2: **Comparison on four-task mixtures.**

| Methods | ChartQA - DocVQA - IconQA - ScienceQA | | | | | | Merge |
|---|---|---|---|---|---|---|---|
| | ChartQA | DocVQA | IconQA | SQA | Avg | Norm | |
| *Mixture of 4 Tasks* | | | | | | | |
| Individual FT | 33.00 | 30.64 | 77.53 | 87.51 | 57.17 | 100 | - |
| LoRA | 32.80 | 30.86 | 73.65 | 85.32 | 55.66 | 97.36 | ✓ |
| DoRA | 32.12 | 30.80 | 73.65 | 85.92 | 55.62 | 97.29 | ✓ |
| CorDA | 31.52 | 29.57 | 72.59 | 86.32 | 55.00 | 96.20 | ✓ |
| DARE | 23.52 | 25.41 | 55.15 | 74.32 | 44.60 | 78.01 | ✓ |
| KnOTS | 23.72 | 24.44 | 56.51 | 75.10 | 44.94 | 78.61 | ✓ |
| HydraLoRA$_{(3B1A)}$ | 32.96 | 30.66 | 73.05 | 84.93 | 55.40 | 96.90 | ✗ |
| HydraLoRA$_{(4B1A)}$ | 33.32 | 30.79 | 72.51 | 84.93 | 55.39 | 96.89 | ✗ |
| ThanoRA | 32.72 | 30.62 | 75.08 | 87.36 | **56.45** | **98.74** | ✓ |

ples low-rank direction and magnitude for improved performance; (d) **CorDA** Yang et al. (2024), which previews task instructions to initialize LoRA modules based on knowledge priors; (e) **DARE** Yu et al. (2024), a drop-and-rescale strategy that sparsifies delta weights for efficient multimodel fusion without retraining or GPU resources; (f) **KnOTS** Stoica et al. (2025), a data-free LoRA merging method that leverages SVD to align task-specific updates. and (g) **HydraLoRA** Tian et al. (2024), an asymmetric MoE-style LoRA framework that excels in handling complex multi-task scenarios. We evaluate its two official settings: 3B1A and 4B1A (i.e., three or four $rank=32$ B matrices with one $rank=32$ A matrix), both with parameter counts no less than standard LoRA with $rank=64$. All other baselines are configured with a default LoRA rank of 64 for fair comparison. For discussions under varying rank settings, please refer to Sec. 4.3.3.

---
[1]https://github.com/haotian-liu/LLaVA

Table 4: **Comparison of Inference Latency (s) and Throughput (it/s)** across datasets. Arrows indicate changes relative to HydraLoRA$_{(3B1A)}$. **_Accuracy_** denotes the average performance across tasks in the two-task mixture setting in Tab. 1.

| Method | ChartQA | DocVQA | IconQA | ScienceQA | CSQA | OBQA | Accuracy |
|---|---|---|---|---|---|---|---|
| _Latency (s) ($\downarrow$)_ | | | | | | | |
| HydraLoRA$_{(3B1A)}$ | 802 | 2095 | 1298 | 723 | 100 | 40 | 63.40 |
| HydraLoRA$_{(4B1A)}$ | 885$_{\uparrow 10.35\%}$ | 2342$_{\uparrow 11.79\%}$ | 1443$_{\uparrow 11.17\%}$ | 785$_{\uparrow 8.58\%}$ | 109$_{\uparrow 9.00\%}$ | 44$_{\uparrow 10.00\%}$ | 63.02 |
| **ThanoRA** | **592**$_{\downarrow 26.18\%}$ | **1516**$_{\downarrow 27.64\%}$ | **1048**$_{\downarrow 19.26\%}$ | **489**$_{\downarrow 32.37\%}$ | **64**$_{\downarrow 36.00\%}$ | **26**$_{\downarrow 35.00\%}$ | **65.08** |
| _Throughput (it/s) ($\uparrow$)_ | | | | | | | |
| HydraLoRA$_{(3B1A)}$ | 3.14 | 2.55 | 4.86 | 5.86 | 12.07 | 12.18 | 63.40 |
| HydraLoRA$_{(4B1A)}$ | 2.84$_{\downarrow 9.55\%}$ | 2.28$_{\downarrow 10.59\%}$ | 4.38$_{\downarrow 9.98\%}$ | 5.40$_{\downarrow 7.85\%}$ | 11.18$_{\downarrow 7.37\%}$ | 10.95$_{\downarrow 10.10\%}$ | 63.02 |
| **ThanoRA** | **4.24**$_{\uparrow 35.03\%}$ | **3.53**$_{\uparrow 38.43\%}$ | **6.03**$_{\uparrow 24.07\%}$ | **8.65**$_{\uparrow 47.61\%}$ | **18.79**$_{\uparrow 55.68\%}$ | **18.71**$_{\uparrow 53.61\%}$ | **65.08** |

Table 5: **Relative performance to _Individual FT_**. Computed as (method / Individual FT) * 100.

| Methods | IconQA - ScienceQA | | | ChartQA - DocVQA | | | CSQA - OBQA | | | Merge |
|---|---|---|---|---|---|---|---|---|---|---|
| | IconQA | ScienceQA | Avg | ChartQA | DocVQA | Avg | CSQA | OBQA | Avg | |
| KnOTS | 90.96 | 96.19 | 93.73 | 83.09 | 92.43 | 87.59 | 97.48 | 92.13 | 94.77 | ✓ |
| HydraLoRA$_{(4B1A)}$ | 99.41 | 100.56 | 100.02 | 98.06 | 101.66 | 99.81 | 98.90 | 99.48 | 99.19 | ✗ |
| **ThanoRA** | **102.00** | **102.55** | **102.29** | **100.61** | **101.93** | **101.26** | **103.30** | **105.25** | **104.29** | ✓ |

## 4.2 COMPARISON TO STATE-OF-THE-ARTS

**Performance Evaluation.** We evaluate `ThanoRA` across diverse multi-task settings, with results reported in Tabs. 1, 2 and 5. Here, "Avg" denotes the average performance across tasks, while "Norm" represents the percentage of Avg relative to individual fine-tuning performance, quantifying how closely multi-task learning approaches the strong baseline individual fine-tuning, following prior work Stoica et al. (2025). Key observations are summarized:

**i) Limited multi-task adaptability of existing methods.** Strong baselines (e.g., CorDA, DoRA) excel on text-only multi-task benchmarks but struggle on more complex multimodal ones; Training-free approaches (e.g., DARE, KnOTS) enable efficient LoRA model merging but still lag behind multi-task training in overall performance.

**ii) ThanoRA robustly improves multi-task adaptation.** ThanoRA consistently achieves superior performance across diverse task combinations and multimodal benchmarks. Moreover, by enabling more effective knowledge injection into low-rank subspaces, it surpasses even single-task fine-tuning, which serves as a strong multi-task training baseline.

Table 3: **Ablation Study of Key Modules.**

| HASI | SPR | IconQA-ScienceQA | | | CSQA-OBQA | | |
|---|---|---|---|---|---|---|---|
| | | IconQA | SQA | Avg | CSQA | OBQA | Avg |
| LoRA | | 77.47 | 87.61 | 82.54 | 74.86 | 78.80 | 76.83 |
| ✓ | | 78.28 | 89.04 | 83.66 | 76.82 | 79.00 | 77.91 |
| ✓ | ✓ | **79.08** | **89.74** | **84.41** | **76.99** | **80.20** | **78.60** |

**Efficiency Analysis.** We compare ThanoRA with the MoE-based method HydraLoRA in terms of both inference efficiency and accuracy, as reported in Tab. 4. Compared with HydraLoRA$_{(3B1A)}$, ThanoRA not only achieves consistently higher accuracy but also reduces latency by **20%–35%** and improves throughput by **24%–56%** across all tasks. The 4B1A variant of HydraLoRA further increases latency and decreases throughput around **10%**, highlighting the scalability limitations of MoE-based LoRA. All evaluations are conducted on a single NVIDIA L20 GPU. Additional results in Appendix D further demonstrate the training efficiency of ThanoRA compared to baselines. These results underscore the practical advantage of mergeable LoRA designs like ThanoRA, particularly in latency-sensitive deployment scenarios such as on edge devices.

Overall, `ThanoRA` delivers a favorable balance between accuracy and efficiency, making it well-suited for real-world multi-task applications with both performance and deployment constraints.

## 4.3 DIAGNOSTIC ANALYSIS

### 4.3.1 ABLATION STUDY.

**Ablation of Key Components.** Tab. 3 shows that HASI alone yields substantial gains over the LoRA baseline, while adding SPR further improves performance by mitigating subspace collapse, confirming their complementary roles.

**Ablation of Hyperparameters.** We evaluate the impact of three key hyperparameters: the regularization weight $\lambda$, scaling factor $\gamma$, and entropy temperature $\tau$. Optimal values ($\lambda=10^{-4}$, $\gamma=5$, $\tau=0.2$) are used in main experiments.

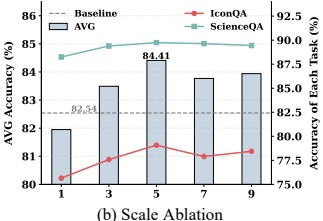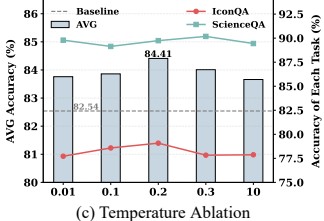

(a) Lambda Ablation      (b) Scale Ablation      (c) Temperature Ablation

Figure 3: **Hyperparameter Study.** Impact of regularization weight $\lambda$, Scale factor $\gamma$, and Temperature $\tau$ on accuracy in IconQA–ScienceQA mixture. ThanoRA consistently outperforms the LoRA baseline (gray dashed line) under almost all configurations. Please see Sec. 4.3.1 for details.

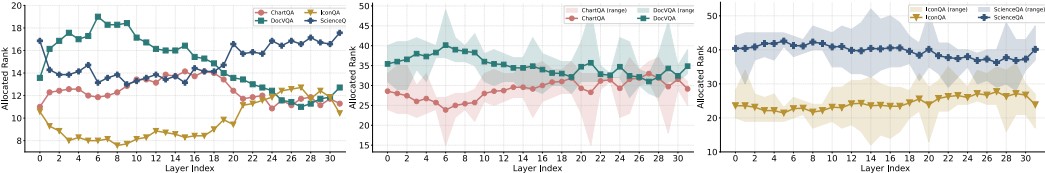

Figure 4: **Layer-wise Rank Allocation.** Lines indicate average allocated ranks per layer; shaded areas show rank range across blocks. Higher ranks in shallow layers for *DocVQA* and in deep layers for *ScienceQA* align with their respective visual and reasoning complexities. Please see Sec. 4.3.2.

### 4.3.2 VISUALIZATION OF RANK ALLOCATION.

Fig. 4 visualizes the layer-wise rank allocation of ThanoRA across task mixtures. In the four-task setting, DocVQA Mathew et al. (2021) receives the highest ranks in shallow layers, while IconQA Lu et al. (2021) consistently obtains the lowest, consistent with the relative complexity of their visual inputs (see Appendix A). This supports prior findings that shallow layers of MLLMs focus on cross-modal fusion Zhang et al. (2024). In contrast, deeper layers allocate higher ranks to ScienceQA Lu et al. (2022); Huang et al. (2025), reflecting its reliance on complex textual reasoning, consistent with the semantic role of later layers.

### 4.3.3 EXPERIMENTS AND ANALYSIS ON DIFFERENT RANKS.

We evaluate the performance of ThanoRA under different LoRA ranks $r \in \{16, 32, 64, 128\}$, as shown in fig. 5. Across all settings, both ThanoRA and its variant without SPR consistently outperform the LoRA baseline, demonstrating its robustness. Notably, vanilla LoRA fails to benefit from increasing rank, indicating inefficient use and redundancy of the parameter space. In contrast, initializing with HASI yields the best performance at $r = 32$, indicating improved rank efficiency through task-aware subspace construction. When further combined with

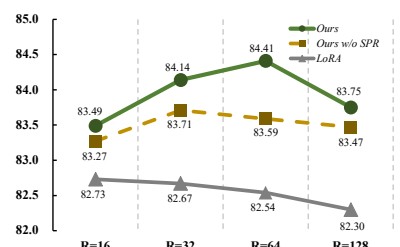

Figure 5: **Performance Under Different Rank.** Please refer to Sec. 4.3.3 for details.

SPR, performance continues to improve and peaks at $r = 64$, confirming that SPR successfully *mitigates subspace collapse* and enables more effective knowledge injection across tasks.

## 5 CONCLUSION

In this paper, we propose `ThanoRA`, an effective and efficient Task Heterogeneity-Aware framework for multi-task low-rank adaptation without non-mergeable structures. By combining *Heterogeneity-Aware Subspace Initialization (HASI)* with *Subspace-Preserving Regularization (SPR)*, ThanoRA enables fine-grained knowledge injection and prevents subspace collapse. Experiments on multi-modal and text-only benchmarks show that it consistently outperforms strong baselines and even surpasses single-task fine-tuning, while maintaining the mergeability of LoRA and thus incurring no additional storage cost or inference latency.

## THE USE OF LARGE LANGUAGE MODELS (LLMS)

We clarify that LLMs were used exclusively for language refinement—improving academic expression, coherence of arguments, and overall clarity. All core ideas, research designs, experiments, and conclusions are solely the work of the authors.

## REPRODUCIBILITY

For reproducibility, the supplementary material includes the source code of the training and evaluation frameworks, while all experimental setups and training hyperparameters are detailed in Sec. 4.1.

## ETHICS AND SOCIAL IMPACT

Our method enables efficient multi-task adaptation without introducing additional non-mergeable structures, preserving both training and inference efficiency. This design makes it particularly suitable for deployment in resource-constrained environments, such as edge devices or low-latency applications. By supporting multiple downstream capabilities with minimal overhead, our approach can facilitate broader accessibility to foundation models in domains like mobile health, assistive technologies, and real-time decision-making.

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

APPENDIX

## A   DATASET DETAILS

Here we provide detailed descriptions of the six downstream datasets used in our experiments. These datasets cover diverse modalities and task types, including both multimodal benchmarks with distinct visual-textual characteristics and text-only reasoning benchmarks, allowing a comprehensive evaluation of multi-task adaptation under heterogeneous settings.

**ScienceQA.**   ScienceQA Lu et al. (2022) is a multimodal multiple-choice benchmark focused on science education, featuring questions grounded in both textual and visual contexts. We adopt 5,000 training and 2,017 test examples. Each sample presents a science question accompanied by both image-based and textual options, and the model is tasked with selecting the correct answer label (e.g., "A", "B"). Accuracy is used as the primary evaluation metric.

**IconQA.**   IconQA Lu et al. (2021) targets abstract diagram understanding, challenging models to perform reasoning over symbolic and schematic visual inputs. We adopt the multiple-choice setting with 5,000 training and 6,316 test instances. Each question requires the model to identify the correct option by outputting the corresponding letter label. Evaluation is based on accuracy.

**ChartQA.**   ChartQA Masry et al. (2022) is a visual question answering benchmark based on bar and line charts, requiring models to interpret and reason over structured chart content. Each instance presents a chart and a natural language question, with the answer being a short free-form textual span (typically a word or phrase). Relaxed Accuracy is used for evaluation, which considers a prediction correct if it approximately matches any reference answer under normalized comparison rules.

**DocVQA.**   DocVQA Mathew et al. (2021) is a visual question answering benchmark on document images, where models must extract and reason over text-rich visual content. Each instance comprises a document image and a natural language question, with the expected output being a short free-form textual answer. ANLS (Average Normalized Levenshtein Similarity) is used as the evaluation metric, measuring string-level similarity between the predicted and reference answers while allowing for minor variations in phrasing or spelling.

**CommonSenseQA (CSQA).**   CommonSenseQA Talmor et al. (2018) is a text-only multiple-choice question answering benchmark designed to evaluate a model's ability to perform commonsense reasoning. Each instance consists of a natural language question and five answer options, where only one is correct. The model is required to return the correct option label. We use 5,000 training and 1,221 test samples. Accuracy is used for evaluation.

**OpenBookQA (OBQA).**   OpenBookQA Mihaylov et al. (2018) is a text-only multiple-choice question answering benchmark designed to assess deeper understanding of scientific topics and language. Inspired by open-book exams, it requires models to combine a small set of provided science facts with broader commonsense and multi-hop reasoning. Each instance consists of a question with four answer options, and the model is expected to select the correct option. The dataset contains 5,000 training and 500 test samples. Accuracy is used as the evaluation metric.

## B   DISCUSSION

Several prior works have explored orthogonality in LoRA to enhance generalization, including PEGO Hu et al. (2024) for domain generalization and O-LoRA Wang et al. (2023) for continual learning. PEGO imposes orthogonality across full LoRA update matrices $\mathbf{W}_t = \mathbf{B}_t \mathbf{A}_t$ to promote feature diversity under distribution shift. Its regularization takes the form:

$$\mathcal{L}_{\text{diversify}} = \sum_{i=1}^{N} \sum_{j=i+1}^{N} \left\| (B_i A_i)^\top (B_j A_j) \right\|_1,  \tag{13}$$

where $N$ is the number of tasks, and $B_i A_i$ denotes the LoRA update for task $i$. However, this formulation incurs substantial computational and memory costs due to full-rank pairwise projections at each layer. As shown in Tab. 6, this design quickly leads to out-of-memory (OOM) errors under modest settings. To address this, we adopt a factorized formulation that independently enforces orthogonality over $\mathbf{A}_t$ and $\mathbf{B}_t$. Crucially, we prove (see Prop. 3.2) that this factor-wise constraint is a sufficient condition for the orthogonality of the resulting LoRA updates $\mathbf{B}_t \mathbf{A}_t$, enabling efficient subspace separation with significantly lower memory overhead, while preserving the training efficiency that is essential for scaling LoRA to larger models and datasets.

O-LoRA adopts an asymmetric regularization strategy by enforcing orthogonality only on the LoRA-A matrices. While this design reduces computational overhead, it is inherently incompatible with our SVD-guided subspace initialization (HASI), which jointly determines the task-specific subspaces through the coordinated initialization of both LoRA-A and LoRA-B matrices. Asymmetric regularization on only one factor disrupts this alignment, undermining the structural integrity established during initialization.

Notably, O-LoRA emphasizes knowledge isolation between LoRA modules across sequential tasks in continual learning. In contrast, our method imposes structural constraints within LoRA by regularizing subspaces across tasks, while still allowing unconstrained components to facilitate inter-task collaboration (see eq. (9)). This distinction reflects different goals: isolating inter-task interference versus enabling coordinated multi-task adaptation within a mergeable framework.

Table 6: **Memory overhead (MB) analysis of LoRA orthogonal regularization.** "QKV Only" denotes applying regularization to LoRA of $W_Q, W_K, W_V$ only; "All Layers" includes all LoRA-injected layers. OOM indicates out-of-memory on single 48GB device. LoRA rank is set to 64.

| Memory Overhead (MB) | ThanoRA | | PEGO Hu et al. (2024) | |
|---|---|---|---|---|
| | QKV Only | All Layers | QKV Only | All Layers |
| 2 Tasks | 65.16 | 206.45 | 15360.23 | OOM |
| 4 Tasks | 237.39 | 751.36 | OOM | OOM |

## C    PROOF OF PROPOSITIONS

### C.1    PROOF OF PROP. 3.1

**Proposition 3.2** (Blockwise composition). *Let* $\mathbf{B}_t^\ell \in \mathbb{R}^{d_{\text{out}} \times r_t}$ *and* $\mathbf{A}_t^\ell \in \mathbb{R}^{r_t \times d_{\text{in}}}$ *for tasks* $t = 1, \ldots, T$, *and define* $\mathbf{B}^\ell = [\mathbf{B}_1^\ell, \ldots, \mathbf{B}_T^\ell] \in \mathbb{R}^{d_{\text{out}} \times r}$ *and* $\mathbf{A}^\ell = [\mathbf{A}_1^\ell; \ldots; \mathbf{A}_T^\ell] \in \mathbb{R}^{r \times d_{\text{in}}}$ *with* $r = \sum_t r_t$. *Then, for any input* $\mathbf{X} \in \mathbb{R}^{d_{\text{in}} \times n}$,

$$\mathbf{B}^\ell \mathbf{A}^\ell \mathbf{X} = \sum_{t=1}^{T} \mathbf{B}_t^\ell \mathbf{A}_t^\ell \mathbf{X}. \tag{14}$$

*Proof.* By block multiplication, we first compute

$$\mathbf{A}^\ell \mathbf{X} = \begin{bmatrix} \mathbf{A}_1^\ell \\ \vdots \\ \mathbf{A}_T^\ell \end{bmatrix} \mathbf{X} = \begin{bmatrix} \mathbf{A}_1^\ell \mathbf{X} \\ \vdots \\ \mathbf{A}_T^\ell \mathbf{X} \end{bmatrix} \in \mathbb{R}^{r \times n}.$$

Left-multiplying by $\mathbf{B}^\ell = [\mathbf{B}_1^\ell, \ldots, \mathbf{B}_T^\ell] \in \mathbb{R}^{d_{\text{out}} \times r}$ yields

$$\mathbf{B}^\ell (\mathbf{A}^\ell \mathbf{X}) = [\mathbf{B}_1^\ell, \ldots, \mathbf{B}_T^\ell] \begin{bmatrix} \mathbf{A}_1^\ell \mathbf{X} \\ \vdots \\ \mathbf{A}_T^\ell \mathbf{X} \end{bmatrix} = \sum_{t=1}^{T} \mathbf{B}_t^\ell (\mathbf{A}_t^\ell \mathbf{X}) = \sum_{t=1}^{T} \mathbf{B}_t^\ell \mathbf{A}_t^\ell \mathbf{X},$$

$\square$

### C.2    PROOF OF PROP. 3.2

**Proposition 3.2** (Sufficient Condition for Orthogonality of LoRA Updates). *Let* $\mathbf{W}_1 = \mathbf{B}_1 \mathbf{A}_1 \in \mathbb{R}^{d_{out} \times d_{in}}$ *and* $\mathbf{W}_2 = \mathbf{B}_2 \mathbf{A}_2 \in \mathbb{R}^{d_{out} \times d_{in}}$. *If either*

$$\mathbf{B}_1^\top \mathbf{B}_2 = \mathbf{0} \quad \text{or} \quad \mathbf{A}_1 \mathbf{A}_2^\top = \mathbf{0}, \tag{15}$$

*then* $\langle \mathbf{W}_1, \mathbf{W}_2 \rangle_F = 0$.

*Proof.* By definition of Frobenius inner product:

$$\langle \mathbf{W}_1, \mathbf{W}_2 \rangle_F = \mathrm{Tr}(\mathbf{W}_1^\top \mathbf{W}_2) = \mathrm{Tr}(\mathbf{A}_1^\top \mathbf{B}_1^\top \mathbf{B}_2 \mathbf{A}_2). \tag{16}$$

We consider two cases:

**(i)** If $\mathbf{B}_1^\top \mathbf{B}_2 = \mathbf{0}$, then:

$$\mathrm{Tr}(\mathbf{A}_1^\top (\mathbf{B}_1^\top \mathbf{B}_2) \mathbf{A}_2) = \mathrm{Tr}(\mathbf{A}_1^\top \cdot \mathbf{0} \cdot \mathbf{A}_2) = 0.$$

**(ii)** If $\mathbf{A}_1 \mathbf{A}_2^\top = \mathbf{0}$, then using the cyclic property of the trace:

$$\mathrm{Tr}((\mathbf{A}_1^\top \mathbf{B}_1^\top)(\mathbf{B}_2 \mathbf{A}_2)) = \mathrm{Tr}(\mathbf{B}_2 \mathbf{A}_2 \mathbf{A}_1^\top \mathbf{B}_1^\top) = \mathrm{Tr}(\mathbf{B}_2 (\mathbf{A}_1 \mathbf{A}_2^\top)^\top \mathbf{B}_1^\top) = \mathrm{Tr}(\mathbf{B}_2 \cdot 0 \cdot \mathbf{B}_1^\top) = 0.$$

Hence, in either case, the Frobenius inner product vanishes. □

## D  TRAINING EFFICIENCY

In this section, we analyze the computational cost of ThanoRA and compare its overall training efficiency with MoE-based counterparts.

### D.1  ONE-TIME INITIALIZATION COST

Following Sec. 3.2.1, ThanoRA requires computing the inverse covariance $\mathbf{C}^{-1}$ and the singular value decomposition (SVD) of $\mathbf{WC}$ to extract task-specific subspaces. Importantly, these computations are performed *only once at initialization* and thus introduce no additional overhead during training. The runtime of these operations is summarized in Tab. 7.

Table 7: One-time initialization runtime.

| Operation | Runtime |
|-----------|---------|
| $\mathbf{C}^{-1}$ | 249s |
| SVD | 708s |

As shown, the total cost is less than 1,000 seconds, which accounts for only **5.1%** of the training time for 3 epochs and merely **1.5%** for 10 epochs. This confirms that the initialization overhead is negligible compared with the overall training process.

### D.2  OVERALL TRAINING TIME AND PERFORMANCE

We further compare ThanoRA with HydraLoRA Tian et al. (2024) in terms of training time and accuracy. Results are summarized in Tab. 8. ThanoRA achieves better performance while requiring less training time, highlighting its advantage in both effectiveness and efficiency.

Table 8: Comparison of total training time and accuracy.

| Method | Training Time | Avg. Accuracy (%) |
|--------|---------------|-------------------|
| HydraLoRA | 6:26:50 | 55.40 |
| ThanoRA | 5:08:22 | 56.45 |

These results demonstrate that ThanoRA not only delivers superior accuracy but also reduces overall training time compared with MoE-based multi-task LoRA methods, underscoring its practical advantage in efficiency-sensitive scenarios.

