# OpenReview forum: "ThanoRA: Task Heterogeneity-Aware Multi-Task Low-Rank Adaptation"
_ICLR.cc/2026/Conference — Submitted to ICLR 2026_

### Official Review · Reviewer_ioF9 · 2025-10-29

**Soundness:** 3
**Presentation:** 2
**Contribution:** 3
**Rating:** 6
**Confidence:** 2

**Summary:**

This paper proposes a novel framework named ThanoRA (Task Heterogeneity-Aware Multi-Task Low-Rank Adaptation), which aims to address the limitations of existing parameter-efficient fine-tuning (PEFT) methods for multi-task learning in large foundation models. Current mainstream approaches, such as Model Merging, can combine tasks without training but often underperform compared to joint multi-task training. Meanwhile, methods based on Mixture of Experts (MoE) can improve performance but introduce non-mergeable routing structures, resulting in additional inference overhead.

ThanoRA achieves multi-task learning by customizing subspace allocation at initialization and enforcing diversity during training. It allocates different dimensions to construct task-specific low-rank subspaces driven by task heterogeneity, thereby enabling fine-grained knowledge injection. Additionally, through an orthogonality regularization term, it encourages the subspaces of different tasks to remain independent while preserving a shared collaborative subspace. This approach mitigates task interference and subspace collapse, fully leveraging the low-rank capacity.

**Strengths:**

- **Clear Motivation and Problem Definition**: The paper excellently dissects the inherent flaws of the two mainstream approaches in existing efficient multi-task fine-tuning methods (Model Merging vs. MoE). Based on this analysis, it presents a clear and valuable research objective: achieving superior multi-task learning performance while maintaining the mergeability of LoRA.
- **Novel and Elegantly Designed Method**: The two core components of ThanoRA, HASI and SPR, are highly innovative. HASI cleverly employs spectral entropy to quantify task complexity and guide hierarchical rank allocation, which is a novel idea. SPR addresses the subspace conflict issue through a computationally efficient decomposed orthogonality constraint, balancing effectiveness and efficiency. The overall methodological framework is designed quite elegantly.

**Weaknesses:**

- **Scalability to Large-Scale Task Numbers**: The experiments were primarily conducted in settings with 2 and 4 mixed tasks. The computational complexity of the SPR regularization term grows quadratically with the number of tasks T ($O(T^2)$). Although the overhead is manageable for a small number of tasks, it may become a training bottleneck when the number of tasks scales up to tens or more. The paper would be more convincing if it discussed the scalability of this method to larger task numbers or proposed potential optimization strategies.
- **Insufficient Analysis of the “Cooperative Subspace”**: The paper introduces a “Cooperative subspace” designed to facilitate knowledge sharing among tasks. However, it lacks in-depth analysis of what specific shared knowledge this subspace learns and how it interacts with task-specific subspaces. For instance, visualization or similarity analysis could be employed to explore its representational characteristics, making the method design more transparent and complete.
- **Hyperparameter Sensitivity and Selection**: The method introduces several key hyperparameters, such as the regularization weight $λ$, scaling factor $γ$, and temperature coefficient $τ$. From the hyperparameter tuning experiments shown in Figure 3, the model's performance exhibits some sensitivity to the values of these hyperparameters. Although the authors provided the optimal values used in the experiments, it would be helpful to offer principles or heuristic methods for selecting these hyperparameters based on task characteristics to enhance the method's practical usability. The sample size used for HASI initialization appears to be empirically chosen. However, in high-dimensional settings, if the number of samples used to estimate the activation covariance matrix is insufficient, the resulting estimates are often dominated by noise, affecting their stability and reliability. The authors are advised to supplement the paper with theoretical analysis or empirical results to support the choice of this sample size.

**Questions:**

- Regarding SPR, its computational overhead increases quadratically with the number of tasks T ($O(T^2)$). Have the authors evaluated the training efficiency and feasibility of this method in scenarios with a larger number of tasks (e.g., $T>10$)? Are there any approximation or optimization methods to reduce this complexity?
- Could the authors provide more analysis or intuition about the role of the “Cooperative subspace”? For instance, what types of shared knowledge does this randomly initialized and unconstrained subspace eventually capture? How does its rank ($r_{coop}$) affect the final performance?
- In Figure 5, the performance of the baseline LoRA slightly decreases as the rank increases (from 16 to 128), which is explained as inefficient utilization of the parameter space. This is an interesting phenomenon. Are there any other optimization-related factors (such as learning rate strategies) that could lead to this result? Besides HASI and SPR, does ThanoRA have any other mechanisms to avoid similar performance degradation at higher ranks?
- HASI relies on a small number of samples (e.g., 500) for “instruction preview” to compute the SVD. Is the quality of this initialization sensitive to the selection of these preview samples? For example, if the distribution of the samples deviates from the entire training set, will it affect the final performance?

---

### Official Review · Reviewer_BNFK · 2025-10-30

**Soundness:** 2
**Presentation:** 2
**Contribution:** 2
**Rating:** 2
**Confidence:** 4

**Summary:**

Current strategies for efficient multi-task setting include model merging and MoE. However, model merging may lead to performance degradation while MoE introduces extra deployment and computation cost. Thus, this paper proposed ThanoRA, which is a two-stage framework to address task heterogeneity. First, they allocate a shared subspace and task-specific subspaces. Second, they propose an orthogonality-aware loss function during training. Extensive experiments and ablation studies are conducted to validate their framework.

**Strengths:**

- The motivation of this paper is clear and good: Model merging is efficient during inference, but may lead to performance degradation, while MoE can maintain performance, but additional modules introduce extra storage and computation cost.
- The experiments are extensive. Multiple ablation studies are conducted to analyze the properties of the proposal.

**Weaknesses:**

- The writing of this paper seems hasty and needs to be improved, and some notations are abused and unclear. Please see the questions.
- The proposed framework relies on some assumptions that I am unsure whether are valid in practice. Moreover, some strategies look too intuitive and unreasonable to me. Please see the questions.
- The scope of this paper is not clear to me and needs further explanation. Please see the questions.

**Questions:**

- In Proposition.3.1, the shape of $B_t^l$ should be $d_{out} \times r_t^l$ instead of $d_{out} \times r_t$?
- In Eq.3, does $\mathbf{\sigma}$ share the same dimension across different tasks and layers? If not, it would be better to have the subscript of layers or tasks. A follow-up question is, if the dimensions are different, is it appropriate to assign $\lfloor \frac{r_{total}}{T+1} \rfloor$ as the shared subspace?
- Why do we use $\lfloor \frac{r_{total}}{T+1} \rfloor$ as the cooperative subspace? Is there any specific reason or motivation? Using an average rank as the shared subspace is confusing to me.
- The initialization in Sec.3.2.1 requires the communication among tasks. This leads to two questions. First, under what situation would this setting be valid? Second, what is the benefit of this work beyond one-shot or few-shot FL, such as [1]?
- Is the rank in truncated SVD in Eq.6 the same as $r_t^l$?
- It seems [2] is related to this paper as they proposed an initialization as well, but the author did not discuss it. The scope of [2] is not that clear either, but maybe the author can rethink their work based on [2].
- Again, Eq.11 requires the communication among clients, which could be multi-round, and the author should discuss the application scenario where this assumption is valid.
- If I understand correctly, during the training phase, we will use the resulting LoRA blocks in Eq.9. If yes, then this will introduce computation or storage overhead. Also, the application scenario should be discussed since the LoRA blocks from all the tasks are kept. Maybe the author should discuss the conventional multi-task setting.
- Though the author studies the inference latency, I think the main cost is in the training phase.
- Not sure if the author will merge the learned LoRA modules after training, though they claim modules can be merged in Fig.1.
- I think DARE is usually used with **other** merging methods, and it is not clear what "DARE" means in Tab.2.

[1] Wang, Ziyao, et al. "One Communication Round is All It Needs for Federated Fine-Tuning Foundation Models." arXiv preprint arXiv:2412.04650 (2024).

[2] Zhang, Haobo, and Jiayu Zhou. "Unraveling LoRA Interference: Orthogonal Subspaces for Robust Model Merging." arXiv preprint arXiv:2505.22934 (2025).

---

### Official Review · Reviewer_5yFn · 2025-10-31

**Soundness:** 2
**Presentation:** 2
**Contribution:** 2
**Rating:** 4
**Confidence:** 4

**Summary:**

This paper proposes ThanoRA, a task-heterogeneity-aware multi-task low-rank adaptation method that leverages spectral-entropy-driven subspace initialization (HASI) and subspace-preserving regularization (SPR) to dynamically allocate LoRA ranks and mitigate subspace collapse, thereby enhancing both performance and stability in multi-task learning.

**Strengths:**

The paper introduces a spectral-entropy-based task complexity modeling method that adaptively allocates LoRA ranks for each task and layer, enabling automatic adjustment of subspace capacity in multi-task settings.

Through rank allocation visualization (Fig. 4) and layer-wise rank distribution across tasks (Fig. 5), the paper demonstrates the model’s ability to automatically align with semantic hierarchies, thereby enhancing the interpretability of the proposed approach.

**Weaknesses:**

	The design of the SPR regularization term remains largely empirical. Although Proposition 3.2 provides sufficient conditions, the paper does not clarify which form of orthogonality (on A or B) is more effective in practice. Moreover, enforcing orthogonality introduces substantial additional training time.

	The ablation study is not sufficiently comprehensive, lacking analyses on the shared cooperative subspace as well as comparisons with alternative rank allocation strategies (e.g., uniform allocation).

**Questions:**

1.	The explanation that the entropy of the singular values of the incremental matrix reflects the task complexity seems somewhat intuitive — could the authors provide a more rigorous or analytical justification?

2.	What are the spectral entropy values for different tasks, and how many ranks are allocated accordingly? A case study or concrete examples would strengthen the analysis.

3.	After computing the required ranks for different tasks, how are these ranks actually applied to initialize the matrices? This implementation detail seems unclear in the paper and is a crucial step in the proposed method.

4.	While orthogonality constraints can encourage diversity in rank representations, how exactly does this mechanism address task heterogeneity?

5.	What are the current state-of-the-art (SoTA) methods being compared against? The description in Section 4.2 appears somewhat inaccurate.

6.	Regarding rank orthogonality, it would be helpful to include additional visualizations to verify subspace separability in the feature space.

---

### Official Review · Reviewer_USGj · 2025-11-01

**Soundness:** 3
**Presentation:** 2
**Contribution:** 2
**Rating:** 4
**Confidence:** 3

**Summary:**

This paper proposes ThanoRA, a task heterogeneity-aware low-rank adaptation framework for efficient and effective multi-task fine-tuning of large models. The method addresses limitations of LoRA-based multi-task adaptation where shared low-rank subspaces cause task interference and performance degradation, and MoE-based LoRA approaches that introduce inference overhead and prevent parameter merging. ThanoRA introduces two key components heterogeneity-aware subspace Initialization and subspace-preserving regularization. Experiments on multimodal and text-only multi-task benchmarks show that ThanoRA outperforms LoRA, DoRA, CorDA, HydraLoRA, and other strong baselines, achieving even higher accuracy than individual fine-tuning while maintaining mergeability and inference efficiency.

**Strengths:**

1. The paper proposes an elegant and practically relevant way to handle task heterogeneity in multi-task LoRA adaptation, combining entropy-based rank allocation  and decomposed orthogonality regularization. The synergy between these two modules is well-motivated and experimentally validated. Unlike MoE-based methods, ThanoRA maintains full mergeability and zero inference overhead.

2. The methodology is mathematically sound and well-justified. The blockwise composition proposition and orthogonality condition are rigorously derived, while computational efficiency considerations demonstrate thoughtful design. Ablations on λ, γ, τ, and rank (r) reinforce the method’s robustness.

3. Strong and consistent performance gains on multi-task benchmarks demonstrate both generality and scalability.

4. Given the increasing industrial use of LoRA for multi-domain model adaptation, a mergeable and heterogeneity-aware multi-task approach without MoE routers is timely and impactful. ThanoRA has potential practical value for efficient adaptation of large language and multimodal models.

**Weaknesses:**

1. The introduced information-theoretic calibration term is central to the method, yet its intuition and derivation could be elaborated further. Specifically, the paper states that the term encourages alignment by minimizing mutual information between task identity and normalized representations, but does not empirically analyze how much task identity leakage remains after optimization. A deeper interpretability or visualization study would strengthen the claim.

2. While ThanoRA presents a well-motivated and empirically effective approach for handling heterogeneous multi-task adaptation, its novelty and theoretical depth are limited when viewed in light of recent advances such as VIB-MTL [1] (VIB+UW) and VIP-MTL [2], which also addresses the heterogeneous multi-task issue. ThanoRA mitigates task heterogeneity primarily through feature-level normalization and an alignment regularizer, but it lacks a probabilistic or variance-invariant formulation that guarantees fairness and balance across tasks.
- [1] Multi-task variational information bottleneck. Arxiv 2020.
- [2] Impartial Multi-task Representation Learning via Variance-invariant Probabilistic Decoding. ACL 2025.

3. Absence of theoretical analysis on fairness or representation balance. While HASI and SPR intuitively mitigate subspace collapse, the paper lacks a formal information-theoretic or variance-decomposition justification showing that task representations remain unbiased or variance-balanced across tasks.

4. Limited exploration of large-scale or highly imbalanced task settings. Experiments are restricted to small-scale mixtures. It remains unclear whether HASI’s entropy-based allocation remains effective for dozens of tasks or severely unbalanced data distributions.

**Questions:**

1. How does ThanoRA handle cases where tasks have vastly different dataset sizes or noise levels? Does entropy-based allocation scale in such imbalanced scenarios?

2. Could you provide quantitative overhead analysis of SPR for larger T (e.g., > 8 tasks)?

3. What advantages does the SPR principle of the proposed method offer compared to balancing multiple heterogeneous tasks through variational methods (i.e., [1], [2], [3])?
- [3] Variational multi-task learning with gumbel-softmax priors. NeurIPS 2021.

4. Could the HASI+SPR framework be extended to other PEFT paradigms (e.g., prefix-tuning or adapter fusion)?

5. Information Content: Have you analyzed whether HASI improves information retention per subspace (e.g., via mutual information or spectral energy retention metrics)?

---

### Meta-Review · Area_Chair_PeNQ · 2025-12-26

**Summary:**

Although the paper targets an important problem, reviewers found that key components are largely heuristic, with limited theoretical justification and unclear novelty relative to prior heterogeneous multi-task learning methods. The evaluation is restricted to small-scale settings, leaving scalability, computational cost, and critical design choices insufficiently validated, and the paper also suffers from clarity and presentation issues. Thus, these concerns prevent the paper from meeting the acceptance bar in its current form.

**Reviewer Concerns:**

Since the authors chose not to participate in the rebuttal, the reviewers’ concerns remain unaddressed.

**Reviewer Scores:**

Since the authors chose not to participate in the rebuttal, the reviewers’ concerns remain unaddressed. As a result, the reviewers are expected to retain their original scores.

---

### Decision · Program_Chairs · 2026-01-26

Reject